# Evaluation of a Care Coordination Initiative in Improving Access to Dental Care for Persons with Disability

**DOI:** 10.3390/ijerph16152753

**Published:** 2019-08-01

**Authors:** Caroline Gondlach, Céline Catteau, Martine Hennequin, Denise Faulks

**Affiliations:** 1General dental practitioner, 42260 Saint Germain Laval, France; 2Réseau Santé Bucco-Dentaire et Handicap de la région Rhône-Alpes (RSBDH), CH le Vinatier, 69000 Lyon, France; 3Faculté de Chirurgie Dentaire, Université Lille 2, 59000 Lille, France; 4Centre for Clinical Research in Dentistry CROC, Université Clermont Auvergne, 63000 Clermont-Ferrand, France; 5CHU Clermont-Ferrand, Service d’Odontologie, 63003 Clermont-Ferrand, France

**Keywords:** disability, oral health, care coordination initiative, access to care, equity of healthcare

## Abstract

In French law, the state is responsible for ensuring equal access to health care for people with disabilities. No system exists within dentistry to guarantee this—there are no salaried public service workers, over 85% of dentists work in general practice, and hospital dentistry is poorly developed. Public funding is available for care coordination initiatives termed “Health Networks”. The objective of this study is to report on an internal evaluation of the Réseau Santé Bucco-Dentaire et Handicap de la région Rhône-Alpes (RSBDH), a Health Network coordinating dentistry for persons with disability in the Rhône-Alpes region, and to discuss the French model of Health Networks as a response to improve access to care. Existing governmental guidelines for the evaluation of Networks were adapted for the RSBDH. The RSBDH coordinated dentists to ensure screening, prevention, and treatment for 3219 persons with disability in 2015. Identified strengths included the identification of vulnerable persons, improved access to treatment and collaboration with primary care services. Weaknesses included training of professionals, continuity of care, information sharing, and stakeholder participation. In 2015, the cost was €501 per patient. This model raises major issues of cost, training, equity, and quality of care within special care dentistry. This discussion is relevant to many countries where models of service provision are currently being developed.

## 1. Introduction

Inequality in oral health is a major problem amongst persons with disability. This population has greater oral health needs than the general population, but lower access to dental services, reflected in high levels of unmet dental treatment need [1,2,3,4,5]. In addition, health care disparity is high. Patients with disabilities who do manage to access services undergo more extractions and receive less conservative or prosthetic treatment than the general population [6]. Public health responses to address these issues accept that it is easier to reduce inequality in access to health care than to reduce inequality in health care outcomes [7]. When conceptualising access to care, Penchansky and Thomas’s model of accessibility, availability, acceptability, affordability, and accommodation (or adequacy) is the most common tool [8]. Contemporary authors have added the domain of awareness/approachability, which relates to the fact that people need to know that a service exists before they can approach it for care [9,10].

In order to address issues of accessibility, accommodation, and awareness, programmes of care coordination have been developed. Care coordination is “the deliberate organisation of patient care activities between two or more participants involved in a patient’s care to facilitate the appropriate delivery of health care services” [11]. In France, these initiatives are often housed within an administrative entity called a “Health Network”, which is financed by the French state [12]. The objective of a Health Network is to improve access to treatment, and to ensure the coordination, continuity, and multidisciplinarity of patient management within a given geographical territory and within a given health domain [13]. In particular, Health Networks are designed to ensure the coordination of local health services for persons with complex medical and/or psycho-social needs. Health Networks are not intended to provide health care but rather to direct patients to appropriate pre-existing public services. The evaluation of Health Networks is mandatory for their funding but is most often internal and usually takes the form of basic end of year reports to the funding regional health authority. National guidelines have been published for the purpose of evaluating Health Networks [14].

French law declares that the state is responsible for compensatory measures to ensure equal access to health care for people with disability [15]. However, no system currently exists within dentistry to guarantee this right—there are no salaried public dental service workers, so dental care is almost exclusively provided within general dental practice, with all the constraints that this small business model implies [16]. In addition, there is no academic specialty in special care dentistry, although a hospital-based clinical specialty called “Medicine Bucco-Dentaire” was created in 2011 [16]. This clinical specialty is aimed at “patients with severe or complex health needs” but depending on the university, this has been interpreted in terms of complex oral health needs, leading to training in advanced acts of treatment within existing academic disciplines (e.g., complex endodontics, maxillofacial prosthodontics) rather than training to care for those with complex general health needs including population groups with intellectual disabilities and the dependent elderly, etc.

One local response to the problem of access to dental care for persons with disability has been the creation of Health Networks. The first disability network for dental treatment was established approximately 15 years ago, and 13 French regions now have a Health Network in this sphere, although these all vary in their target populations, structure, and financing. The Health Network for disability and oral health of the former Rhône-Alpes region (RSBDH—Réseau Santé Bucco-Dentaire et Handicap) exists within its current form since 2013. It was created in response to the lack of dental services available to persons with disability in the region, with aid and funding from the regional health authority. The RSBDH is, to the authors’ knowledge, the first dental network to have undergone formal internal evaluation using national guidelines.

The objective of this manuscript is to describe the results of the internal evaluation of the RSBDH and to discuss the French model of Health Networks as a response to improve access to dental care for persons with disability.

## 2. Materials and Methods

### 2.1. The RSBDH Health Network

The RSBDH covers the whole of the former Rhône-Alpes region, representing a population of 6.45 million inhabitants in 2014 (the French regions were reorganised in 2017). Its stated objectives are: To coordinate health care professionals to ensure screening, early intervention, treatment, and prevention of oral disease.To improve the individual and collective competence of the Health Network’s actors and partners and, as a result, to improve the quality and the continuity of care for patients.To improve the effectiveness and the efficiency of patient management.To describe the target population and evaluate procedures (for example special care techniques and pain management).

Patients are eligible to join the RSBDH if they present an intellectual disability with behavioural difficulties, cerebral palsy with severe neurological impairment, multiple disabilities, autism or neurodevelopmental disorders on the autistic spectrum, rare disorders associated with intellectual disability, or if they are dependent elderly with behavioural issues (e.g., dementia). Persons with dental anxiety or phobia alone, i.e., with no additional disability over and above anxiety, are excluded from the Network. There are no age restrictions, but beneficiaries must reside within the geographical area of the RSBDH. In addition, socio-medical institutions may pay to register with the Network, which then coordinates all the dental treatment needs of residents and provides annual screening via a mobile dental unit. Institutions must provide annual evidence of oral health promotion activity or oral health education for their staff and residents in order to maintain their registration with the Network. Institutional care and education is still the norm in France for persons with developmental disabilities, often as day-care in children and then residential care in adulthood.

The RSBDH is primarily funded by the regional health authority, with some income received through institutional registration fees and donations. The Network coordinator assesses new patient demands and orientates the patient towards: (1) a local general dental surgery that is a member of the Network; (2) a surgery that is financed by the Network with access to an appropriate clinical environment (presence of a dental assistant, nitrous oxide sedation, and a Network general practitioner); or (3) towards a specialist hospital consultation. Practitioners within the Network can also refer patients for treatment under general anaesthesia at local public or private hospitals. General dental practitioners that register with RSBDH are compensated financially with a session fee, either in addition to the standard social security system fee-per-item tariff, or as a salary. Treatment provided within the Network is limited to extractions and preventive and restorative care, although some prosthetic work is available in certain situations. Orthodontics and implants are excluded from the offer of care. Patients pay the national health service fee for their treatment. France has a fee-per-item system where 70% of the state fixed fee is paid by the national health system and the remaining 30% is paid either by the patient or by complementary health insurance [16]. For a restricted number of medical conditions, the national health service pays the full fee and there is no cost to the patient. Conscious sedation techniques are not, as yet, reimbursed by the national health service and the Network sets a fixed tariff for this modality which is paid by the patient. Treatment under general anaesthesia is free at the point of service.

### 2.2. Tools Used for the Evaluation of the RSBDH

National Health Authority guidelines were published in 2015 to guide the evaluation of practice and organisation within Health Networks [17] and these were followed in the present study. This internal evaluation consists of three parts. The first two evaluate roles that the Network is presumed to fulfil: (1) Support given to professionals for the organisation of care pathways (nine dimensions), and (2) support given for multidisciplinary work and territorial networking (eight dimensions). The third part is concerned with (3) the efficiency of internal organisation (four dimensions). These 21 dimensions are designed to evaluate the Health Network’s ability to improve patient management within their given domain. The guidelines are designed to cover all the different types of Health Networks in all medico-social domains. Sixteen dimensions from this document were applicable to the specific context of the RSBDH and these items were retained (the sixteen dimensions are listed in Table 1). Those dimensions that were excluded related to the wider medical or social context, for example prevention of avoidable repeated admission to hospital, prevention of burn-out of carers, or abuse of patients by carers. In some cases, the wording of the dimensions was adapted to a dental rather than a medical context. Two supporting documents give details as to the use of the guidelines [18,19]. These list the points that should be reported for each dimension and that are scored as “completed”/“partly completed”/“not done”/“not applicable”. As an example, dimension 1 concerns “Targeting of vulnerable persons” and the first point scored is “All stakeholders are involved in the definition of those situations that should be considered as high priority by the Network”.

As the 2015 document does not include any form of economic assessment, previous guidelines from 2004 were used for economic evaluation in the current study [20]. The economic chapter of these guidelines covers the areas of financial resources, analysis of budget, and economic evaluation of action taken. These items are descriptive and are not equivalent to a full economic analysis.

### 2.3. Method for the Evaluation of the RSBDH

A member of the RSBDH board undertook the internal evaluation of the Network, with the help of a steering committee, in collaboration with the Network coordinator, and with the academic support of the Centre for Clinical Research in Dentistry of the University Clermont Auvergne. Data from 2013 to 2015 were obtained and analysed. The founding documents of the Network, the Network statutes and rules of procedure were consulted, as were all minutes of meetings of the Board and Executive Board, the general assemblies, end-of-year reports, and financial statements. In addition, all documents pertaining to the recruitment of patients, of dental practitioners, of institutions, and the website were searched. Additional documents, such as the specifications for the website and patient database, the organisational flowchart etc. were also consulted. These elements were used to complete the 16 dimensions of the internal evaluation as above and to inform the economic assessment.

The expected outcome of the evaluation using the tools described above, was to identify the strengths and weaknesses of RSBDH and to estimate the cost of the Network per patient.

## 3. Results

### 3.1. Overview of the RSBDH

Table 2 describes the turnover of the RSBDH in terms of number of patients, number of registered institutions, number of employees, number of dentists etc. The number of both patients and dentists registered with the Network grew in 2015. In 2015, there was a ratio of 214 “active” patients (those with clinical contact within the year) to one full-time employee. When comparing patient numbers from 2014 with those of the last census in 2014, the Network touched 0.05% of the total population of the region. This may be contrasted with the fact that 10.4% of the population of the region was found to have a functional disability in 2007 [21].

### 3.2. Organisational Evaluation

An overview of the results of the internal organisational evaluation is given in Table 1, with the 16 dimensions retained from the 2015 Health Authority guidelines [17,18,19]. Five of these dimensions were identified as strengths of the Network. In the domain of “Support given to professionals for the organisation of care pathways”, those dimensions for which the majority of items were fulfilled by the RSBDH were: “Targeting of vulnerable persons” and “Providing help to guarantee access to treatment and social assistance”. In the domain of “Support given for multidisciplinary work and territorial networking” the dimension of “Supporting preventive and health promotion activities” was fulfilled. In the third domain, “Efficiency of internal organisation” two out of four dimensions were reached—“Working in collaboration with primary care services” and “Management of resources within the Health Network”.

Weaknesses were identified in the organisation of care pathways, notably in the planning, follow up and re-evaluation of personal oral health care plans; in the orientation of patients with complex needs; and in ensuring continuity of care between primary and secondary health care sectors. All but one dimension of the chapter on multidisciplinary work and territorial networking needed improvement, in particular providing guidance to practitioners in terms of protocols, providing training, sharing of information and working on health and safety aspects of care. In addition, no means of patient feedback were in place.

It was concluded that, whilst the Network does fulfil its stated objective of providing coordination of health care professionals to ensure screening, early intervention, treatment and prevention of oral disease for persons with disability, improvement was needed in the areas of training, continuity of care, information sharing, stakeholder participation, and health and safety issues.

### 3.3. Economic Assessment

The RSBDH received €1,625,000 in 2015, of which 88% came from regional government funds, 6% from institutional registration fees, 1% from donations, 1% from investment and 4% from the back payment of treatment provided by salaried Network practitioners within the hospital setting. In terms of spending, 66% of the budget was spent on treatment and screening, 18% on coordination of care, and 16% on administration. The cost per patient registered with the Network per year in 2015 was €501, over and above the cost of treatment to the national health service, to the patient, and to complementary health insurance plans.

## 4. Discussion

The RSBDH was able to coordinate oral health care professionals regionally to ensure screening, preventive intervention and dental treatment for 3219 persons with disability in 2015. A large minority of these patients were in institutions that registered with the Network (38%). A major limitation as to the generalisability of these results is that the population reached by the Network is so small as to be almost anecdotal, compared to the population of the zone covered (0.05% of the total population). This is in itself revelatory however, as no other system of positive discrimination exists to aid persons with disability to gain access to dental services in the area. It is extremely likely that patient-led demand for services is low, given the lack of visibility of the problem of access to care for this population. No data are available as to the number of persons with disability treated in general dental practice in France. A British estimate suggests that 9 out of 10 persons with disability could potentially be treated in general dental practice if practitioners had appropriate training and other barriers, such as wheelchair access, were addressed [22,23]. However, unpublished data from a pilot study of patient satisfaction within the Network reported that 12 out of 28 respondents had being looking for a dental health care provider for over a year before finding the RSBDH, and that seven of these had been searching for over five years (unpublished data available from corresponding author). There is no reason to believe that access to primary dental care is any better in France than in other European countries, particularly in the absence of systematic teaching of Special Care Dentistry in dental universities [16].

The model of using a care coordination initiative to improve access to general dental services for those with disability is interesting [24]. Care coordination can only address the problem of access if all elements of the model are considered, including accessibility, availability, acceptability, affordability, accommodation (or adequacy), and awareness [8,9,10]. It would seem that the “awareness” or “approachability” dimension of this model is the most lacking in terms of the RSBDH. Improvement here would involve being better aware of local population need in terms of both type and volume. It would also mean better communication in order to ensure that all potential users were aware that the Network existed, what services were offered for whom, and how to enter the system. It is likely that this aspect has been restrained by financial constraints at the health authority level, but also by aspects of service availability, as recruitment of general dental practitioners has been problematic for the Network. When considering affordability, the model seems costly when reduced to the number of beneficiaries, but data are not available to describe the potential cost of non-intervention. It would seem that evaluating the cost-benefit ratio of care coordination interventions is often very difficult, but costs are generally high [25,26].

If the conceptual model for equity in health is taken into account–that people in equal need should have equal access, equal treatment and equal treatment outcomes [27], care coordination only addresses the first part of this equation [28]. The Network presumes competence in the general dental practitioners that register with it and provides little in the way of guidance or protocols. In addition, certain treatment options are excluded from the Network “package”. Aspects of evaluation that address disparities in health care and quality of care should therefore be included in future evaluations [29]. Quality of care and the assurance of equal treatment outcomes is primordial [27]. Patients with special needs have been repeatedly shown to undergo more extractions than the general population [6], despite the fact that these patients have additional difficulties adapting to dentures, if they are provided or feasible. Indeed, the main limitations of this study were inherent in the tools used for the evaluation. Although national guidelines were followed, these guidelines did not conform to theoretical frameworks for the study of care coordination [30,31,32]. Of the 14 key concepts identified by Van Houdt et al., only six were considered within the evaluation proposed [30].

As an initiative to improve access to dental care for persons with disability, the model of Health Networks as structured in France provides only a limited and local response to the problem. Widespread consultation with patient groups and representatives is essential to ensure that the services offered are the services that the population requests and that at least the same level of care is provided as for other populations. This implies developing primary care services that are local, inclusive, person-centred, comprehensive, sufficient in quantity and in quality, and affordable [33,34,35]. It is obvious that efforts should be directed to improving knowledge and attitudes within the primary care arena, such that universal access becomes reality [6,23]. In the current environment however, it may be that the most important roles of Networks, or care coordination initiatives, is in the identification of general dental practitioners with a special interest in special care dentistry and in the provision of a mechanism for remuneration that reflects expertise and time taken in the domain. In addition, Networks may work proactively, by providing screening in an institutional environment for example, in order to respond to reduced expressed need and low patient demand within these populations [23,35,36]. This Network initiative would have to be extended nationally however, with accompanying funding and professional training programmes, before a population-level impact could be evident.

## 5. Conclusions

The RSBDH Network has improved access to dental treatment and prevention for persons with disability within its region. However, this model of care coordination raises major issues of cost, training, equity, and quality of care assurance within special care dentistry. This discussion is relevant to the many countries where models of service provision are currently being developed.

## Figures and Tables

**Table 1 ijerph-16-02753-t001:** Points achieved in each dimension of the internal evaluation of the Réseau Santé Bucco-Dentaire et Handicap de la region Rhône-Alpes (RSBDH) defined by the National Health Authority Guidelines, 2015 [17,18,19].

	Total Number of Points for Each Dimension	Completed	Partly Completed	Not Done	Not Applicable
**PART 1 – Support given to professionals for the organisation of care pathways**					
**Dimension 1 Targeting of vulnerable persons**	5	✔✔✔✔✔			
Dimension 2 Planning, follow-up and re-evaluation of personal oral health care plans	5		✔	✖✖✖	✕
Dimension 3 Orientation of patients with complex needs to appropriate care	5	✔✔	✔✔	✖	
**Dimension 4 Providing help to guarantee access to treatment and social assistance**	5	✔✔✔		✖	✕
Dimension 5 Ensuring continuity of care between primary and secondary health care sectors	5	✔	✔✔✔	✖	
**PART 2 – Support given for multidisciplinary work and territorial networking**					
Dimension 6 Providing guidelines and follow up for multidisciplinary protocols	5			✖	✕✕✕✕
Dimension 7 Supporting action to improve practice and organisation	5	✔✔	✔	✖✖	
**Dimension 8 Supporting preventive and health promotion activities**	5	✔✔✔✔✔			
Dimension 9 Supporting the development of a culture of security of care	5			✖✖✖	✕✕
Dimension 10 Sharing of information	5	✔✔		✖✖	✕
Dimension 11 Providing feedback to the funding body	5	✔	✔	✖	✕✕
Dimension 12 Taking account of the views of the target population	5			✖	✕✕✕✕
**PART 3 – Efficiency of internal organisation**					
**Dimension 13 Working closely with primary health care services**	7	✔✔✔✔✔✔		✖	
**Dimension 14 Managing resources within the Network**	5	✔✔✔		✖	✕
Dimension 15 Ensuring the safety of actions within the Network	5			✖✖✖✖	✕
Dimension 16 Choice and use of appropriate software systems	5	✔✔		✖	✕✕

**Table 2 ijerph-16-02753-t002:** Overview of the Réseau Santé Bucco-Dentaire et Handicap de la région Rhône-Alpes (RSBDH) in figures.

	2013	2014	2015
Number of registered patients	3360	3496	4164
Number of patients with a clinical contact within the year = “Active patients”	2322	2433	3219
Total number of dentists registered	16	24	39
of which the number of general dental practitioners was	9	19	35
Proportion of dentists registered with the Network out of the total number of dentists in the region	unknown	unknown	0.8%
Number of medico-social institutions or care homes registered with the Network	85	74	85
Number of hits on the Network website	45,769	47,070	58,913
Number of equivalent full-time employees	11	16	15
Number of active patients/number of equivalent full-time employees	206	152	215
Number of institutions benefiting from the mobile dental unit	unknown	50	50
Number of patients seen in the mobile dental unit	700	540	530

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
