# Peer review of "Evaluation of a Care Coordination Initiative in Improving Access to Dental Care for Persons with Disability"

_ijerph, 2019, doi:10.3390/ijerph16152753_

Round 1

Reviewer 1 Report

Thank you for the opportunity to review “Effectiveness of care coordination initiative in improving access to dental care for persons with disability”. This manuscript deals with an important topic, and will be of interest to readers. However, there are some issues that need to be addressed before considering it for publication.

Introduction

1.     I would suggest to emphasize the main purpose of your study and to state it from the title, abstract and introduction. There are many dates, but the reader might be a little bit confused.

2.     It is unclear the context and creation of RSBDH. Please, specify this term because it may be confusing for readers.

Material and methods

3.     Please offer a more accurate description of the evaluation of the RSBDH. Please, clarify this information and specify the 21 dimensions and their evaluation. In the manuscript this information is confused.

Results

4. Please, review the instructions of the journal about the format of the tables, so both tables will have the same format.

5. What are the sociodemographic characteristics of the sample? It would be interesting to include a table with that information.

6.     More information about the psychometric properties of the measures used in the study is needed. As reliability varies in each test administration, it is necessary to report the reliability

7.     Please, indicate the sampling method and inclusion criteria.

Discussion

8. In the discussion, is necessary the inclusion of more references.  

9. When you say “Quality of care and the assurance of equal treatment outcomes is primordial” is necessary to add biography.

9. More extensive and detailed information about clinical and practical implications extracted from the study should be included.  

10. Finally, authors are aware that their research was carried out in “France, but only a limited and local response”. But, what happens with another countries or areas? Could be yours results generalizable? A statement about it is also necessary.

Author Response

Please see attached Word document

Reviewer 2 Report

The study entitled “Effectiveness of a care coordination initiative in improving access to dental

care for persons with disability” whose objective was to describe the results of the internal

evaluation of the RSBDH and to discuss the French model of Health Networks as a response to

improve access to dental care for persons with disability. The study has been conducted well

and the manuscript is well presented. However, authors should address some issues before

the manuscript will be considered again for publication.

Introduction

- It is unclear if RSBDH is one of the Health Networks that were created as a local response to

the problem of access to dental care for persons with disability or is in other context. Please

specify. Also, in the text is indicated “The first disability network for dental treatment was

established approximately 15 years ago.”. In the Materials and Methods says RSBDH was

created in 2003. So it seems that RSBDH is was created before Heatlh Networds.

- The introduction section is quite complete and contains theoretical background. Moreover,

their subsections provide an adequate structure. However, more references are needed

especially in the statements where the citations are missed.

Material and methods

- Who evaluates the RSBDH? It is important to clarify it in the text.

- Why the 21 dimensions of the RSBDH were not evaluated? In the paper you say that only 16

dimensions were evaluated. Please, clarify this information. In general, it could be

recommendable extend the information about the evaluation of the RSBDH.

Results

- Tables. Please review the instructions of the journal about the format of the tables. Adapt the

format of the tables to the standards, so both tables will have the same format. add in the end

of the table 1 the meaning of “ND”.

- Table 1. Add at the end of the table 1 the meaning of “ND”. The fourth line is in the right

hand. Also, the 2015 “Number of hits on the Network website” is unreadable.

- In the text “This may be contrasted with the fact that 10.4% of the population of 165 the

region was found to have a functional disability in 2007 [19].” could be necessary update the

data to the period of the study.

Discussion

- In the introduction you say “The Health Network for disability and oral health of the former

Rhône-Alpes region (RSBDH) is, to the authors’ knowledge, the first dental network to have

undergone formal internal evaluation using national guidelines”. Therefore, in the discussion

section could be recommendable debate about the importance of the evaluation of

interventions programs, and about the needed of creating a standard of evaluation.

- In the introduction you say “Persons with dental anxiety or phobia alone are excluded from

the Network.” It could be a bias in your results. For this reason, you should add a statement in

the discussion section to inform readers about this limitation.

Author Response

Please see attached Word document
